# Biomolecular Classification in Endometrial Cancer: Onset, Evolution, and Further Perspectives: A Critical Review

**DOI:** 10.3390/cancers16172959

**Published:** 2024-08-25

**Authors:** Valentina Bruno, Martina Betti, Jessica Mauro, Alessandro Buda, Enrico Vizza

**Affiliations:** 1Gynecologic Oncology Unit, Department of Experimental Clinical Oncology, Istituto di Ricovero e Cura a Carattere Scientifico Regina Elena National Cancer Institute, 00128 Rome, Italy; valentina.bruno@ifo.it (V.B.); enrico.vizza@ifo.it (E.V.); 2Biostatistics, Bioinformatics and Clinical Trial Center, Istituto di Ricovero e Cura a Carattere Scientifico Regina Elena National Cancer Institute, 00128 Rome, Italy; martina.betti@ifo.it; 3Department of Computer, Control, and Management Engineering, University of Rome La Sapienza, 00185 Rome, Italy; 4Division of Gynecologic Oncology, Michele and Pietro Ferrero Hospital, 12060 Verduno, Italy; jessica.mauro@me.com

**Keywords:** biomolecular markers, endometrial cancer, risk subgroups

## Abstract

**Simple Summary:**

This review challenges the current ESGO/ESTRO/ESP guidelines on endometrial cancer risk classification, emphasizing the need to have more extensive and differentiated patient cohorts and to carefully consider the length of follow-up evaluation. Since the new guidelines for endometrial cancer risk classification have been released, many reviews have proposed a critical re-evaluation. In this review, we look back to how the molecular classification system has been built and its evolution in time.

**Abstract:**

Since the new guidelines for endometrial cancer risk classification have been published, many reviews have proposed a critical re-evaluation. In this review, we look back to how the molecular classification system was built and its evolution in time to highlight the major flaws, particularly the biases stemming from the inherent limitations of the cohorts involved in the discoveries. A significant drawback in some cohorts is the inclusion criteria, as well as the retrospective nature and the notably sparse numbers, especially in the *POLEmut* (nonsynonymous mutation in EDM domain of POLE) risk groups, all of which impact the reliability of outcomes. Additionally, a disregard for variations in follow-up duration leads to a non-negligible bias, which raises a substantial concern in data interpretation and guideline applicability. Finally, according to the results that we obtained through a re-analysis of the *confirmation cohort*, the *p53abn* (IHC positive for p53 protein) subgroup, which is predominant in non-endometrioid histology (73–80%), loses its predictivity power in the endometrioid cohort of patients. The exclusion of non-endometrioid subtypes from the cohort led to a complete overlap of three molecular subgroups (all except *POLEmut*) for both overall and progression-free survival outcomes, suggesting the need for a more histotype-specific approach. In conclusion, this review challenges the current ESGO/ESTRO/ESP guidelines on endometrial cancer risk classification and highlights the limitations that must be addressed to better guide the clinical decision-making process.

## 1. Introduction

The article from Levine and The Cancer Genome Atlas Research Network discussed new molecular insights to better sub-classify endometrial cancer patients by recurrence risk for suitable follow-up and adjuvant treatment [1]. For the first time, an integrated molecular analysis has been proposed, consolidating various studies on endometrial cancer genomics previously limited by heterogeneous features to include different subtypes. This study was built on an unsupervised clustering analysis, which accounted for copy-number alteration and mutation-rate profiles of patients from all histotypes. The authors identified four clusters and tried to associate the most representative mutated genes for each group, meaning that the identified genes are “surrogate” of more complex profiles that involve the whole genome. The authors further demonstrated that the “surrogate” profiles, provided by the mutational status of a reduced set of genes, were still able to determine a patient’s outcome, in terms of both overall and progression-free survival.

In 2015, Stelloo et al. proposed using molecular analysis based on “surrogates” of the four original clusters to redefine risk assessment and tailor adjuvant therapy for high-risk endometrial cancer. This approach includes factors such as age, grade, and stage as part of the TransPORTEC [2] study, an international consortium related to the PORTEC3 trial [3]. In this explorative study, molecular analyses were used to identify four prognostic subgroups in terms of survival outcomes and distant metastasis localization: p53 immunohistochemistry, microsatellite instability, POLE proofreading mutation, and no specific molecular profile (NSMP). The authors concluded that this sub-classification might have potential therapeutic implications by the identification of molecular targets for individualized therapy.

From 2015 to 2018, the ProMisE group studies have been published, which refer each to a different dataset: discovery, confirmation, and validation cohorts. In the first study by Talhouk in 2015 [4], the group described the development of the EC molecular classifier through a discovery cohort. In this study, The Cancer Genome Atlas (TCGA) data were used to develop more feasible and cheaper surrogate assays compared to genomic analysis for TCGA classification: combinations of the identified assays were tested on a new independent cohort of 152 EC. The authors concluded by proposing a classifier based on mismatch repair protein immunohistochemistry, POLE mutational analysis, and p53 immunohistochemistry as a surrogate for copy-number (CN) status. Later, L1CAM expression level, in association with p53, has been proposed as a further stratification candidate [5,6]. Evidence suggested that these new factors had an independent prognostic role beyond the already well-known risk factors. In the second study by Talhouk in 2017 [7], the group confirmed the decision-tree classification of ProMisE (Proactive Molecular Risk Classifier for Endometrial Cancer) through a confirmation cohort. In this study, immunohistochemistry (IHC) for the presence or absence of mismatch repair (MMR) proteins, sequencing for polymerase-E (POLE) exonuclease domain mutations [8], and IHC for p53 (*p53wt* and *p53abn*) were carried out on 319 new EC samples to classify patients in the four different previously detected subgroups and to correlate them with their relative outcomes. The ProMisE was then compared with the ESMO (European Society of Medical Oncology) risk-stratification algorithm, showing a better prediction ability in oncological outcomes. In the third study by Kommoss in 2018 [9], the group validated the ProMisE (Proactive Molecular Risk Classifier for Endometrial Cancer) through a validation cohort of 452 women in which clinical and molecular data were analyzed. The authors demonstrated that ProMisE has a high prognostic value as a molecular classification tool in identifying EC prognostic molecular subtypes.

Accordingly, the most relevant society for endometrial cancer care guidelines, started to include in their documents the potential application of the biomolecular classification. The first societies were the European Society for Medical Oncology (ESMO), the European SocieTy for Radiotherapy & Oncology (ESTRO) and the European Society of Gynaecological Oncology (ESGO) joined in a consensus conference on endometrial cancer held in 2014 [10]. Thereafter, a study by the Gynaecologic Oncology Group member institutions revealed many critical points of all non-prospective studies published in Endometrial Cancer (EC) [11], such as the high incidence of serous histotypes in the Copy Number high (*CN-high*) group and the highest rate of treatment administered to POLE-mutated patients as a result of greater incidences of higher grades in this hypermutated subgroup. In addition, this study proposes an alternative class to p53abn, namely the Loss of Heterozygosity (LoH) group, which appears to be more representative in cohorts with a lower proportion of stage I cases.

In 2020, the revision of the 2014 evidence-based guidelines from the European Society of Gynaecological Oncology (ESGO), the European Society for Radiotherapy and Oncology (ESTRO) and the European Society of Pathology (ESP), proposed the molecular classification as a “prognostic” tool encouraged in all endometrial carcinomas to better define prognosis and better address patients to adjuvant treatment [12]. Recently, in June 2023, the International Federation of Gynaecology and Obstetrics (FIGO) Women’s Cancer Committee officially released an updated staging system in which the endometrial cancer biomolecular classification is included [13]. Based on this new evidence [14], current endometrial cancer classification has integrated the conventional morphologic features (such as histopathologic type, grade, myometrial invasion, and LVSI) with molecular classification to provide an additional level of information for better addressing adjuvant treatment in early stages. 

However, over the years many have highlighted the limitations both in data and strategy approaches [15,16,17], and since the new guidelines for endometrial cancer risk classification have been released, many reviews have proposed a critical re-evaluation [18]. In this review, we will critically revise all studies which have contributed to the development of the EC biomolecular classification by highlighting main biases and criticisms.

## 2. Materials and Methods

The parameters that we have taken under consideration for the critical evaluation of these studies are described in the next section.

### 2.1. Literature and Search Strategy

A comprehensive systematic research of all references cited in the “Definition of prognostic risk groups integrating molecular markers” of the ESGO/ESTRO/ESP guidelines [12] for those studies that present original datasets linked to EC biomolecular classification development was performed. The reference lists regarding studies, reviews, and meta-analysis included in the reference guidelines described above were mainly considered (Figure 1). Moreover, two authors (M.B. and V.B.) independently read and specifically evaluated the databases that each manuscript was referring to and performed a more in-depth quality assessment of the retrieved datasets. Four large cohorts have been identified that had a numerosity of over 100 patients (Figure 2). This threshold was set to guarantee a minimum representativity of the minority classes (for instance, *POLEmut* class has an incidence <10%; therefore, in the context of biomolecular classification, it would define a class represented by a handful of patients).

### 2.2. Outcome Measures

#### 2.2.1. Histotype Specificity

To assess the predictive role of each molecular subclass, some considerations will be made on the real impact of such profiles in a histotype-specific manner to avoid redundancy with risk factors that were already considered. To this aim, we re-analyzed the *discovery cohort* presented in 2015 which provides open access to processed data.

#### 2.2.2. Follow-Up

The clinical outcome recorded for the cohorts under review will be evaluated based on the exclusion/inclusion of patients that have been lost at follow-up within two years and on the percentage of *censored* patients within the first five years for each molecular class.

#### 2.2.3. Molecular Profiles Definition and Numerosity

In all the reviewed studies, all patients have been stratified under the molecular profiles. The definitions of the four classes themselves and the decision-tree model employed for the classification will be summarized. The numerosity of each subclass will also be considered to perform a robustness evaluation.

#### 2.2.4. Study Design

The last parameter under examination will be the nature of the study itself, as for strong changes in the clinical guidelines, an adequate design of a prospective study should be required. Moreover, the role of the administration of adjuvant treatments will also be taken into consideration.

## 3. Results

### 3.1. Histotype Specificity

The *p53abn* profile was present with a low percentage of the endometrioid histotype across all considered studies (13–25%), even if this histotype represents the majority of endometrial cancer cases [10] (Figure 2—*p53abn* histology). Consequently, in FIGO 2023 classification *p53* status could be redundant taken together with the histotype. In fact, the *p53* prognostic role has never been shown to be non-redundant with the tumor histotype. This lack of specificity affects all other aspects that we will discuss below since it poses a methodological bias in how study cohorts should be defined. To corroborate these hypotheses, we re-generated the *confirmation* cohort’s Kaplan–Meier (KM) survival curves [7] and demonstrated that, excluding endometrioid cases, the biomolecular risk stratification is no longer significant (*p*-value > 0.05) to define prognosis both in terms of overall survival (OS) and progression-free survival (PFS) (Figure 3). Moreover, we can observe that the Micro Satellite Instability (*MSI)*, *CNlow,* and *p53abn* groups overlap in terms of overall survival and partially in terms of progression risk, both showing a 25% probability of relapse and death within the endometrioid subtype. These results suggest that these three subtypes mostly share prognoses due to their correlation with the non-endometrioid histotype, rather than providing additional information, corroborating our hypothesis that the exclusion of the non-endometrioid subtype from EC datasets is crucial to highlight the real added value of biomolecular classification with respect to standard histological evaluation.

### 3.2. Follow-Up

The follow-up time has been set for a minimum of two years for the majority of datasets (Figure 2). However, the analysis of the KM curves of survival showed that a greater rate of events was observed after two years [10]; therefore, this threshold should be extended for a more meaningful indication of the patient’s outcome. An increase in follow-up period homogeneity should be required to guarantee the same probability of observing an event (relapse/death). The incidence of censored events strongly varies across biomolecular classes, for instance, in the *discovery* and in the *validation* cohorts, the incidence of shorter follow-up times among patients who do not experience relapse/death events is much higher than in other groups (Figure 3). This has represented a non-negligible bias, especially for the *p53abn* profiles in contrast to *MSI, POLEmut,* and *p53wt,* and especially for PFS, which poses a substantial concern in data interpretation and guideline applicability (Figure 3).

### 3.3. Molecular Profiles Definition and Numerosity

The first large study that proposed the molecular characterization of endometrial cancer patients [1] was based on an unsupervised clustering analysis, which accounted for copy-number alteration and mutation-rate profiles of patients from all histotypes. The authors identified four clusters and tried to identify those genes for which mutational status may act as a “surrogate” of more complex profiles. They further demonstrated that the “surrogate” profiles, provided by the mutational status of a reduced set of genes, were still able to discriminate between patient outcomes, in terms of both overall and progression-free survival. However, the re-analysis performed on the *confirmation cohort*, in which non-endometrioid histotypes were excluded, shows that the informativity of molecular classes merely represents the genomic mutation rate (TMB), which is a well-known but hardly specific biomarker (Figure 3A,B). For the purpose of data availability, the same analysis could not be repeated for all datasets, but the proportion of *p53abn* patients within the non-endometrioid subtype would suggest that this aspect remains unchanged in all subsequent cohorts. Moreover, the four molecular profiles have been modified many times (Figure 2) in favor of diagnostic feasibility but to the detriment of accuracy [1]. Moreover, the ordering of the features to be considered in the risk-decision model has also been evaluated and modified multiple times, especially within the *confirmation cohort*. However, it has been shown that, regardless of the order, this model alone is not adequate to achieve the predictive power of the 2009 standard guidelines [4]. This leads to a non-negligible inconsistency of molecular category definition in the studies under review.

In addition, while the overall numerosity has been increasing across studies (Figure 2), when we consider the singular biomolecular subgroup, the *POLEmut* class (around 10% of the cases [19]), seems to be highly underrepresented in all cohorts. Furthermore, if we were to consider, as we propose, the exclusion of the non-endometrioid histological profile, the numerosity of the *p53abn* class would be scarce as well (around 10% of the total number of EC endometrioid patients) (Figure 1). Therefore, if we take together *POLEmut* and *p53abn* profiles within the endometrioid subtype (the predominant EC subtype in terms of incidence), these studies only consider 20% of the entire endometrioid cohort. In turn, this also implies that this molecular characterization can only improve the risk-class assignment for approximately 20% of all EC patients. Considering all of this, we propose that a minority of EC molecular profiles (POLE and p53 status) cannot legitimately justify the modification to the FIGO 2023 guidelines that, in contrast, stand for the entire EC population.

### 3.4. Study Design

The molecular classification has been introduced as a prognostic factor in the 2009 ESGO risk classification guidelines, which are the guidelines used for the patients of the studies. The retrospective nature of the considered studies implies a heterogeneity bias in the applied follow-up schemes and treatment assignments according to the specific biomolecular subgroup. Therefore, these studies designed to demonstrate differences in prognosis according to the molecular subgroups fail to do so since they are themselves influenced by the molecular risk class initially assigned: for instance, the *POLEmut* subgroup shows a significantly higher incidence of incomplete follow-up periods across all studies (Figure 4A,B).

## 4. Discussion

This review challenges the current ESMO/ESGO/ESTRO and the ESGO/ESTRO/ESP guidelines on endometrial cancer risk classification and highlights the necessity for larger and more diverse patient cohorts, along with meticulous consideration of follow-up durations to guarantee an added value to pre-existing risk assessment strategies. The rationale for these statements will be summarized in three main arguments:

Argument 1: The *p53mut* cases are mostly patients with a non-endometrioid subtype, which is well known to have a poor outcome regardless of the molecular profile. At the same time, the *POLEmut* profile is described by a limited number of patients, some of whom have a partial or non-complete follow-up. FIGO 2023 introduced biomolecular classes previously defined by the ESGO-ESTRO-ESP that arise from a minority of EC profiles (*POLEmut* and *p53abn*) that have been extended to all EC populations. 

Argument 2: Due to the retrospective design of studies that led to biomolecular classification, are we convinced that it is possible to not be biased from the inevitable heterogeneity in follow-up schemes and above all treatment assignments conditioned by the risk classes themselves? In this contest, the biomolecular class definition represents both a starting- and an endpoint.

Argument 3: The TCGA biomolecular stratification of EC patients is based on unsupervised clustering of genomic profiles for which some diagnostic-feasible mutational profiles have been selected as surrogates. The author of the study highlighted how these surrogates are not perfect, with *p53abn* having the lowest accuracy across the four classes. Additionally, all studies mentioned in this work have based their risk classification on a decision-tree approach; therefore, the definition of *p53abn* derived from these studies does not coincide with how it has been integrated into the ESMO guidelines. In considering this, should we not try to generate better surrogates for the originally derived risk classes? According to our considerations and other reviews [18], the molecular profiles defined as they are today no longer provide a sufficient level of informativity for patient-risk stratification [20].

## 5. Conclusions

The results of this review provide evidence underline the necessity for larger and more diversified patient cohorts and meticulous consideration of follow-up durations. Most importantly, a re-evaluation of molecular markers should be made to move to a more histotype-specific approach to ensure added value over pre-existing risk assessment strategies.

## Figures and Tables

**Figure 1 cancers-16-02959-f001:**
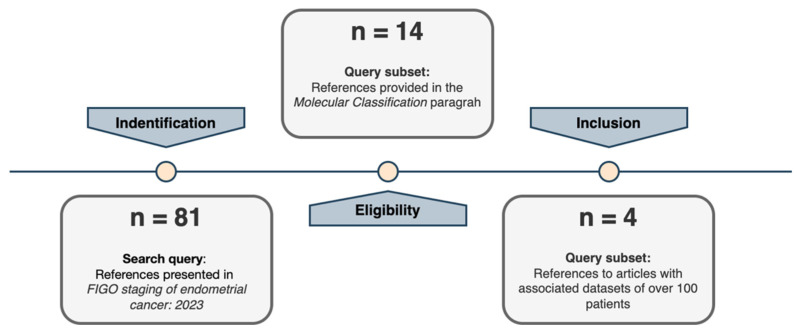
Selection study steps flow chart.

**Figure 2 cancers-16-02959-f002:**
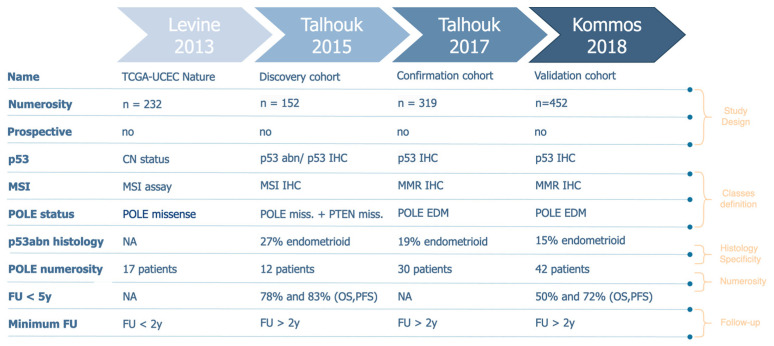
Characteristics of the selected databases.

**Figure 3 cancers-16-02959-f003:**
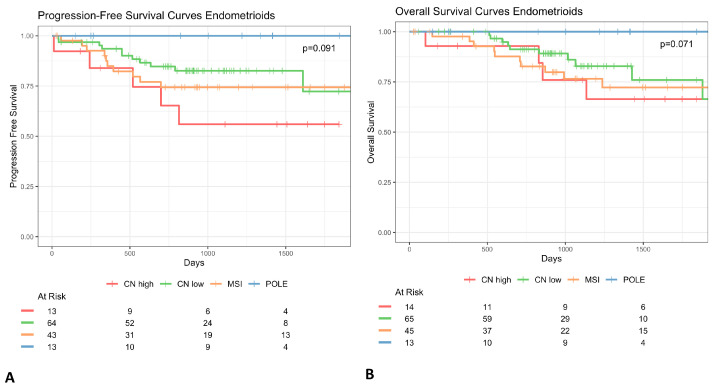
Kaplan–Meier survival curves of progression-free survival (**A**) and overall (**B**) of women with non-endometrioid endometrial cancer. Survival curves show that biomolecular risk stratification appears no longer significant for the prognosis (*p* value > 0.05).

**Figure 4 cancers-16-02959-f004:**
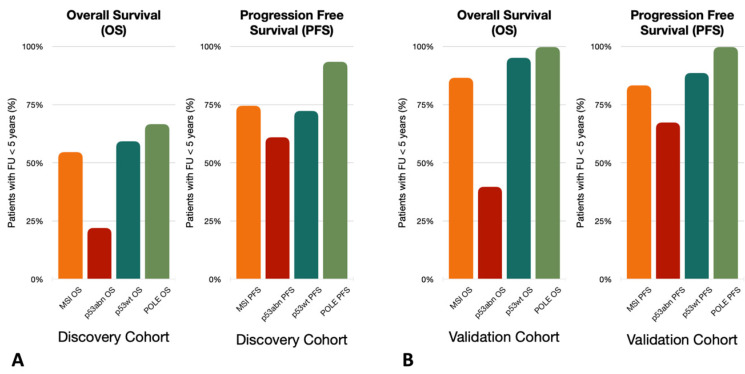
The incidence of follow-up in less than 5 years across patients for which the event of interest, overall survival (OS) and progression-free survival (PFS) was not experienced. (**A**) Discovery Cohort incidence of follow-up in less than 5 years across patients for which the event of interest (OS and PFS) was not experienced. (**B**) Validation Cohort incidence of follow-up in less than 5 years across patients for which the event of interest (OS and PFS) was not experienced.

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
