# Peer review of "Biomolecular Classification in Endometrial Cancer: Onset, Evolution, and Further Perspectives: A Critical Review"

_cancers, 2024, doi:10.3390/cancers16172959_

Round 1

Reviewer 1 Report

Comments and Suggestions for Authors

Please find my comments below:

  • The study does not offer substantial new contributions or insights beyond what is already available in the current literature.
  • The retrospective nature of the studies reviewed introduces significant biases, which compromise the reliability of the findings and conclusions.
  • Sample sizes for the risk groups are insufficient, weakening the validity of the analysis.
  • The emphasis on molecular subgroups, such as POLEmut and p53abn, is problematic because these subgroups represent a small fraction of endometrial cancer cases, limiting the broader applicability of the findings.
  • Histotype specificity is not adequately considered; including non-endometrioid histotypes dilutes the significance of the prognostic value of the biomolecular risk stratification.
  • Follow-up periods vary significantly across studies, leading to inconsistencies and potential biases in the outcomes.
  • The study design and methodology lack robustness, particularly due to the heterogeneity in follow-up protocols and treatment assignments.
  • The proposed biomolecular classification does not significantly outperform existing risk assessment strategies in terms of predictive power.
  • The rationale for revising current guidelines is not convincingly supported by the reviewed data.
  • The review does not adequately address the methodological limitations and biases of the included studies.
  • The suggestions for re-evaluating molecular markers do not offer clear, actionable improvements over existing classifications.
  • The conclusions are not supported by a robust, prospective data set, limiting their applicability in clinical settings.
  • The study calls for larger and more diverse patient cohorts but does not provide substantial new data or methods to achieve this goal.
  • More comprehensive and prospective studies are necessary to validate the findings and recommendations made in this review.

Author Response

  1. The study does not offer substantial new contributions or insights beyond what is already available in the current literature.

Thank you for your comment. To our knowledge, no other studies have presented a list of limitations of studies on which the biomolecular classification is based at the dataset level. If you are referring to specific studies, we would appreciate being redirected to those to improve our claim of novelty.

  1. The retrospective nature of the studies reviewed introduces significant biases, which compromise the reliability of the findings and conclusions.
  2. Sample sizes for the risk groups are insufficient, weakening the validity of the analysis.
  3. The emphasis on molecular subgroups, such as POLEmut and p53abn, is problematic because these subgroups represent a small fraction of endometrial cancer cases, limiting the broader applicability of the findings.
  4. Histotype specificity is not adequately considered; including non-endometrioid histotypes dilutes the significance of the prognostic value of the biomolecular risk stratification.
  5. Follow-up periods vary significantly across studies, leading to inconsistencies and potential biases in the outcomes.
  6. The study design and methodology lack robustness, particularly due to the heterogeneity in follow-up protocols and treatment assignments.
  7. The proposed biomolecular classification does not significantly outperform existing risk assessment strategies in terms of predictive power.
  8. The rationale for revising current guidelines is not convincingly supported by the reviewed data.
  9. The review does not adequately address the methodological limitations and biases of the included studies.

Thank you for raising this points; however, all the limitations mentioned are indeed the limitations of the actual biomolecular classification that our review has pointed out, hence they do not reflect biases in our review itself.

  1. The proposed biomolecular classification does not significantly outperform existing risk assessment strategies in terms of predictive power.

Thank you for your observation. Since this is a review, the authors do not aim to offer actionable improvements but rather encourage the reconsideration of current guidelines due to the inadequacy of the studies on which such guidelines are based.

  1. The conclusions are not supported by a robust, prospective data set, limiting their applicability in clinical settings.

Thank you for your comment. As our aim is to demonstrate the inadequacy of studies on which current guidelines are based, no prospective datasets are required in a review form. This review provides information that could be used to build new RCTs to address issues such as histotype specificity and stronger representation of minority classes.

Comment to editor:

We respectfully believe that this reviewer may not have fully read our article, as his queries do not seem pertinent to the aim of our review, which is a critical revision of studies on which current guidelines are based. Additionally, we suspect that the reviewer comments may have been automatically generated through Large Language Models (e.g., ChatGPT, Gemini). We provide references to tools that support our suspicion: Quillbot AI Content Detector, Scribbr AI Detector.

Reviewer 2 Report

Comments and Suggestions for Authors

The main idea of this Review is that most of the available endometrial cancer risk classification studies are inadequate due to their retrospective nature and limited sample size. This inadequacy is getting worse by the inclusion of patients with inconsistent follow-up periods, which leads to unreliable results and leads to errors in the validity of the data and the applicability of recommendations

This suggests a need to re-evaluate molecular markers towards a more histotype-specific measure for the current risk assessment strategy. Addressing these limitations is critical for governing bodies to better support effective decision making.

This review is distinguished by its original approach, clarity and relevance. It can be published in Cancers.

 At the same time, there are several small points:

1. Abstract: Please, explain the aim of the Review, describe the most important results and underline principally novel analytical findings

2. It also would be good to uncover abbreviation "MSI-high, p53abn, and p53wt" in the Abstract.

3. Introduction: Please uncover abbreviation where it is appropriate.

4. Conclusion:  Underline novelty of the Review in more details.

Author Response

The main idea of this Review is that most of the available endometrial cancer risk classification studies are inadequate due to their retrospective nature and limited sample size. This inadequacy is getting worse by the inclusion of patients with inconsistent follow-up periods, which leads to unreliable results and leads to errors in the validity of the data and the applicability of recommendations

This suggests a need to re-evaluate molecular markers towards a more histotype-specific measure for the current risk assessment strategy. Addressing these limitations is critical for governing bodies to better support effective decision making.

This review is distinguished by its original approach, clarity and relevance. It can be published in Cancers. 

At the same time, there are several small points: 

  1. Abstract: Please, explain the aim of the Review, describe the most important results and underline principally novel analytical findings
  2. Done
  3. It also would be good to uncover abbreviation "MSI-high, p53abn, and p53wt" in the Abstract. 
  4. done
  5. Introduction: Please uncover abbreviation where it is appropriate.
  6. done
  7. Conclusion:  Underline novelty of the Review in more details. 
  8. done

We thank the reviewer for appreciating the goal of our review and for highlighting what was unclear in the manuscript. We have added a list of abbreviations section and explained the molecular classes at their first appearance. In the abstract, we now use full terms. Furthermore, on request of other reviewers, we integrated results with discussion to make the aim more clear in the conclusions and we believe the key points now appear more clear.

Reviewer 3 Report

Comments and Suggestions for Authors

This study critically appraises the implementation of biomolecular classifications in endometrial cancer, with a particular reference to the limited specificity of the p53abn profile across its histotypes, as well as its implication in prognostic stratification. This underlies the methodological bias in the realization and reporting of retrospective cohort studies, recommending a remedy for a re-review of the integration of biomolecular markers with traditional histological classifications to enhance patient risk stratification and indications for treatment. A few comments below are considered by the authors:

Introduction:

1. Acronyms, such as FIGO, TCGA, and ProMisE, are commonly introduced within the text without their full form.

2. Failure to present a critical analysis and commentary on the limitations associated with the cited studies.

3. Break down lengthy sentences into shorter, more manageable ones. for example but not limitted ""For the first time an integrated molecular analysis has been proposed to bring together all the different previously published studies on genomics in endometrial cancer, which all were limited by heterogeneous histological and classification features, to build a more consolidated knowledge by including different subtypes" ; "A comprehensive systematic research of all references cited in the 'Definition of prognostic risk groups integrating molecular markers' of the ESGO/ESTRO/ESP guidelines (9) for those studies that present original datasets linked to endometrial cancer biomolecular classification development""

4. The different methodologies used in these studies were compared and contrasted.

methods:

5. Terms like "discrepancies were resolved by consensus" need more context about how consensus was achieved.

6.  Some of the criteria applied remain unexplained in terms of logic; for example, the exclusion of studies with fewer than 100 patients.

7. The importance of each parameter in the context of the endometrial cancer classification can be emphasized.

8. The discussion of Kaplan-Meier survival curves and the importance of biomolecular risk stratification would be greatly improved by a more thorough description and interpretation of the results.

9. Example for Improvement: "To confirm these hypotheses, we re-generated the confirmation cohort’s Kaplan

 Survival curves (5) ascertained similar results, stating that, except for endometrioid cases, the biomolecular risk stratification was no longer significant, both in terms of overall and progression-free survival, to specify prognosis, that is, p value >0.05 (Figure 3). This could be expanded to include individual data points or actual numbers, but for clarity alone.

 10. A more detailed explanation of the overlap of MSI, CNlow, and p53abn groups in the survival results would be good.

Author Response

This study critically appraises the implementation of biomolecular classifications in endometrial cancer, with a particular reference to the limited specificity of the p53abn profile across its histotypes, as well as its implication in prognostic stratification. This underlies the methodological bias in the realization and reporting of retrospective cohort studies, recommending a remedy for a re-review of the integration of biomolecular markers with traditional histological classifications to enhance patient risk stratification and indications for treatment. A few comments below are considered by the authors:

Introduction:

  1. Acronyms, such as FIGO, TCGA, and ProMisE, are commonly introduced within the text without their full form.

Thank you for pointing this out. All abbreviations have been indicated in their full form at their first appearance as per journal guidelines and are now listed in a list of abbreviations.

  1. Failure to present a critical analysis and commentary on the limitations associated with the cited studies.

Thank you for your feedback. Our aim was not to criticize the scientific approach, rationale, and rigor of the cited studies but rather the criteria used for dataset definition and generation. By addressing the limitations that characterize the datasets, we implicitly highlight the limitations of the conclusions reached in these studies.

  1. Break down lengthy sentences into shorter, more manageable ones. for example but not limitted ""For the first time an integrated molecular analysis has been proposed to bring together all the different previously published studies on genomics in endometrial cancer, which all were limited by heterogeneous histological and classification features, to build a more consolidated knowledge by including different subtypes" ; "A comprehensive systematic research of all references cited in the 'Definition of prognostic risk groups integrating molecular markers' of the ESGO/ESTRO/ESP guidelines (9) for those studies that present original datasets linked to endometrial cancer biomolecular classification development""

Three long sentences have been rephrased or shortened. They now read:

“Line 36: The article from Levine and The Cancer Genome Atlas Research Network discussed new molecular insights to better sub-classify endometrial cancer patients by recurrence risk for suitable follow-up and adjuvant treatment

Line 39: For the first time, an integrated molecular analysis has been proposed, consolidating various studies on endometrial cancer genomics, previously limited by heterogeneous features, to include different subtypes

Line 47: In 2015, Stelloo et al. proposed using molecular analysis based on "surrogates" of the four original clusters to redefine risk assessment and tailor adjuvant therapy for high-risk endometrial cancer. This approach includes factors like age, grade, and stage as part of the TransPORTEC study, an international consortium related to the PORTEC3 trial.”

  1. The different methodologies used in these studies were compared and contrasted.

We strongly agree with the reviewer, since it is one of the major limitations arising from the methodologies comparison of analyzed studies, that we have described in the “outcome measure” paragraph.

  1. Terms like "discrepancies were resolved by consensus" need more context about how consensus was achieved.

Currently, this sentence does not appear in the manuscript.

  1. Some of the criteria applied remain unexplained in terms of logic; for example, the exclusion of studies with fewer than 100 patients.

Thank you for highlighting this. The “Literature Search and Search Strategy” has been modified to clarify our criteria, ensuring a minimum representativity of the minority classes. It now reads:

“A comprehensive systematic research of all references cited in the “Definition of prognostic risk groups integrating molecular markers” of the ESGO/ESTRO/ESP guidelines (9) for those studies that present original datasets linked to EC biomolecular classification development. The reference lists regarding studies, reviews, and meta-analysis included in the reference guidelines described above, were mainly considered (Figure 1). Moreover, two authors (M.B. and V.B.) independently read and specifically evaluated the databases that each manuscript was referring to, in order to perform a more in-depth quality assessment of the retrieved datasets. Four large cohorts have been identified that had a numerosity of over 100 patients. This threshold was set to guarantee a minimum representativity of the minority classes (for instance, POLEmut class has an incidence <10%, therefore in the context of biomolecular classification it would not make sense to define a class representated by a handful of patients).”

  1. The importance of each parameter in the context of the endometrial cancer classification can be emphasized.

The importance of features and their clinical impact on endometrial cancer management has been now provided in the introduction section:

“On the basis on these new evidences [15], current endometrial cancer classification has integrated the conventional morphologic features (such as histopathologic type, grade, myometrial invasion, and LVSI) with the molecular classification, to provide an additional level of information for better addressing adjuvant treatment.”

  1. The discussion of Kaplan-Meier survival curves and the importance of biomolecular risk stratification would be greatly improved by a more thorough description and interpretation of the results.

  1. Example for Improvement: "To confirm these hypotheses, we re-generated the confirmation cohort’s Kaplan Survival curves (5) ascertained similar results, stating that, except for endometrioid cases, the biomolecular risk stratification was no longer significant, both in terms of overall and progression-free survival, to specify prognosis, that is, p value >0.05 (Figure 3). This could be expanded to include individual data points or actual numbers, but for clarity alone.

  1. A more detailed explanation of the overlap of MSI, CNlow, and p53abn groups in the survival results would be good.

Further details have been added into the manuscript which now reads:

“To corroborate these hypotheses, we re-generated the confirmation cohort’s Kaplan-Meier (KM) survival curves (5) and demonstrated that, excluding endometrioid cases, the biomolecular risk stratification is no longer significant (pvalue > 0.05) to define prognosis both in terms of  Overall Survival (OS) and Progression Free Survival (PFS) (Figure 2). Moreover, we can observe that the Micro Satellite Instability (MSI), CNlow and p53abn groups overlap in terms of overall survival and partially in terms of progression risk, both showing a 25% probability of relapse and death within the endometrioid subtype. These results suggest that these three subtypes mostly discriminate prognosis due to their correlation with the non-endometrioid histotype rather than providing additional information, corroborating our hypothesis that the exclusion of the non-endometrioid subtype from EC datasets is crucial to highlight the real added value of the biomolecular classification with respect to standard histological evaluation. “

Reviewer 4 Report

Comments and Suggestions for Authors

The study demonstrates that biomolecular classification is important in endometrial cancer.

The objective of the study is very important, however, the text, discussion and references are insufficient for a full review paper. It may be better to change the type of the paper into short communication or some kind of mini-review or survey. Otherwise, expansion of the texts and references to review the biomolecular classification in endometrial cancer.

1. The main question addressed by the research is to review the molecular classification on endometrial cancer
2. The study is original in terms of describing outcome measures.
3. Characteristics of the selected databases in Figure 2 is unique, however, the relationship between molecular markers is not clear enough.
4. It has been described that a total of 81 potential articles were identified and 14 studies included molecular classification in Methods. These 14 studies may be cited as references.
5. Conclusion is unclear whether molecular markers have been identified in the review study. The type of the article is supposed to be a review article.
6. A number of references is insufficient for a review article.
7. Figure 4 may be revised to add the source of data in a reference reviewed.

Author Response

The study demonstrates that biomolecular classification is important in endometrial cancer.

The objective of the study is very important, however, the text, discussion and references are insufficient for a full review paper. It may be better to change the type of the paper into short communication or some kind of mini-review or survey. Otherwise, expansion of the texts and references to review the biomolecular classification in endometrial cancer.

1. The main question addressed by the research is to review the molecular classification on endometrial cancer

  1. The study is original in terms of describing outcome measures.

  2. Characteristics of the selected databases in Figure 2 is unique, however, the relationship between molecular markers is not clear enough.

Thank you for your feedback. Figure 2 does not aim to directly describe parameters related to molecular markers; instead, it aims to describe parameters related to the datasets from which the markers have been identified as prognostic features.

  1. It has been described that a total of 81 potential articles were identified and 14 studies included molecular classification in Methods. These 14 studies may be cited as references.

References for all 14 studies emerged from literature search have been inserted throughout the manuscript and they are all in the bibliography. Additional references from the consensus conferences have also been added.

  1. Conclusion is unclear whether molecular markers have been identified in the review study. The type of the article is supposed to be a review article.

We did not identify or aim to identify molecular markers in this review. The reanalysis provided in the studies only wants to highlight the criticism associated with the lack of histotype specificity to derive statistically significant results.

  1. A number of references is insufficient for a review article.

We are aware of the limited number of studies presented, however since we aim to discuss current guidelines on molecular classification, we were limited by the number of studies mentioned within the official guidelines document.

  1. Figure 4 may be revised to add the source of data in a reference reviewed.

We added references for each reported cohort within the figure 4 legend.

Reviewer 5 Report

Comments and Suggestions for Authors

The review critically evaluates the current limitations in biomolecular classification of endometrial cancer, particularly focusing on the inadequacy of retrospective studies, the need for homogeneous follow-up times, and the reevaluation of molecular markers within specific histological subtypes. It emphasizes the importance of adopting a more histotype-specific approach to improve risk assessment and clinical decision-making, suggesting future research directions to refine and enhance current guidelines. The topic holds significance in the oncology field, however, there are major knowledge gaps and missing points which needs to be included. My comments are given below-

The authors have structured the paper in a research article style, classifying content into distinct results and discussion sections. To avoid confusion and enhance clarity for readers, it would be beneficial to modify the format to align more closely with a review or meta-analysis style. This would involve synthesizing the findings into a cohesive narrative, integrating the results with the discussion to provide a more comprehensive and accessible overview of the current state of biomolecular classification in endometrial cancer.

The review correctly highlights the inadequacy of many retrospective studies in endometrial cancer research. Retrospective analyses often suffer from selection bias and lack of standardized treatment protocols, which can significantly impact the validity and generalizability of the findings. The paper points out the notably sparse numbers in specific risk groups such as POLEmut. This underrepresentation limits the statistical power and reliability of conclusions drawn from these cohorts, emphasizing the need for larger, more comprehensive studies.

The inclusion of patients with inadequate and non-homogeneous follow-up times is a significant drawback. This heterogeneity can lead to skewed outcomes and misinterpretations, underlining the importance of consistent and adequate follow-up periods in clinical studies. The review discusses the disregard for variations in follow-up duration, which introduces bias. This is a critical point, as inconsistent follow-up can lead to erroneous conclusions about the efficacy and prognosis associated with different biomolecular subgroups. In section 4. Discussion, with the sentence ´The second concern is related to the rationale used for the biomolecular classification´, cite DOI: https://doi.org/10.37349/ebmx.2024.00009 a recent report on regulatory based molecular profiling requirement in cancer metanalysis.

The finding that the p53abn subgroup loses its predictive power in the endometrioid cohort, as opposed to its strong association with non-endometrioid histology, is significant. This suggests that molecular markers may not be universally applicable across different histological subtypes and need context-specific validation. The review’s observation that excluding non-endometrioid subtypes leads to an overlap of the MSI-high, p53abn, and p53wt subgroups for survival outcomes is noteworthy. This overlap suggests that current molecular classifications may not adequately capture the prognostic nuances within endometrioid endometrial cancer.

The recommendation to reassess molecular markers to adopt a more histotype-specific approach is crucial. This tailored strategy could enhance the precision of risk assessment and treatment planning, potentially improving patient outcomes. Addressing the limitations discussed, such as study design flaws and follow-up inconsistencies, is pivotal for refining clinical guidelines. More robust and standardized research methodologies are needed to provide clearer, evidence-based directives for clinical decision-making.

The paper emphasizes the importance of including homogeneous cohorts in research to reduce bias. Ensuring patient groups are comparable in terms of treatment, follow-up, and baseline characteristics is essential for producing reliable and applicable results. The review identifies several key areas for future research, including larger, prospective studies and the development of more nuanced molecular classifications. This forward-looking perspective is valuable for guiding the next steps in endometrial cancer research. Cite latest report https://doi.org/10.1002/aisy.202300366 with the sentence ending with ´…..such profiles in a histotype specific manner to avoid redundancy with risk factors that were already considered´

By pointing out the current research gaps and biases, the review underscores the need for improved clinical decision-making tools. Better molecular markers and risk classifications will enable more personalized and effective treatment strategies for endometrial cancer patients. The review's reproduction of analysis for the confirmation cohort adds robustness to its findings. Reproducing and validating previous research is a critical step in confirming the reliability of biomolecular classifications and their clinical relevance.

Highlighting the substantial concern of bias in data interpretation due to study limitations, the review calls for a more critical evaluation of existing research. This scrutiny is necessary to advance the field and ensure that clinical practices are based on sound evidence. The discussion on the evolution of biomolecular classification in endometrial cancer is insightful. Understanding how classifications have developed over time helps contextualize current challenges and directs efforts toward more effective future classifications.

Comments on the Quality of English Language

Minor editing of English language required

Author Response

  1. The review critically evaluates the current limitations in biomolecular classification of endometrial cancer, particularly focusing on the inadequacy of retrospective studies, the need for homogeneous follow-up times, and the reevaluation of molecular markers within specific histological subtypes. It emphasizes the importance of adopting a more histotype-specific approach to improve risk assessment and clinical decision-making, suggesting future research directions to refine and enhance current guidelines. The topic holds significance in the oncology field, however, there are major knowledge gaps and missing points which needs to be included. My comments are given below

1.The authors have structured the paper in a research article style, classifying content into distinct results and discussion sections. To avoid confusion and enhance clarity for readers, it would be beneficial to modify the format to align more closely with a review or meta-analysis style. This would involve synthesizing the findings into a cohesive narrative, integrating the results with the discussion to provide a more comprehensive and accessible overview of the current state of biomolecular classification in endometrial cancer.

We thank the reviewer for the thorough description of the main points and for the useful comment on the review structure. The discussion is now included in the results paragraph, while conclusions have been kept separate.

  1. The review correctly highlights the inadequacy of many retrospective studies in endometrial cancer research. Retrospective analyses often suffer from selection bias and lack of standardized treatment protocols, which can significantly impact the validity and generalizability of the findings. The paper points out the notably sparse numbers in specific risk groups such as POLEmut. This underrepresentation limits the statistical power and reliability of conclusions drawn from these cohorts, emphasizing the need for larger, more comprehensive studies.The inclusion of patients with inadequate and non-homogeneous follow-up times is a significant drawback. This heterogeneity can lead to skewed outcomes and misinterpretations, underlining the importance of consistent and adequate follow-up periods in clinical studies. The review discusses the disregard for variations in follow-up duration, which introduces bias. This is a critical point, as inconsistent follow-up can lead to erroneous conclusions about the efficacy and prognosis associated with different biomolecular subgroups. In section 4. Discussion, with the sentence ´The second concern is related to the rationale used for the biomolecular classification´, cite DOI: https://doi.org/10.37349/ebmx.2024.00009 a recent report on regulatory based molecular profiling requirement in cancer metanalysis. 

The finding that the p53abn subgroup loses its predictive power in the endometrioid cohort, as opposed to its strong association with non-endometrioid histology, is significant. This suggests that molecular markers may not be universally applicable across different histological subtypes and need context-specific validation. The review’s observation that excluding non-endometrioid subtypes leads to an overlap of the MSI-high, p53abn, and p53wt subgroups for survival outcomes is noteworthy. This overlap suggests that current molecular classifications may not adequately capture the prognostic nuances within endometrioid endometrial cancer. The recommendation to reassess molecular markers to adopt a more histotype-specific approach is crucial. This tailored strategy could enhance the precision of risk assessment and treatment planning, potentially improving patient outcomes. Addressing the limitations discussed, such as study design flaws and follow-up inconsistencies, is pivotal for refining clinical guidelines. More robust and standardized research methodologies are needed to provide clearer, evidence-based directives for clinical decision-making. The paper emphasizes the importance of including homogeneous cohorts in research to reduce bias. Ensuring patient groups are comparable in terms of treatment, follow-up, and baseline characteristics is essential for producing reliable and applicable results. The review identifies several key areas for future research, including larger, prospective studies and the development of more nuanced molecular classifications. This forward-looking perspective is valuable for guiding the next steps in endometrial cancer research. Cite latest report https://doi.org/10.1002/aisy.202300366 with the sentence ending with ´…..such profiles in a histotype specific manner to avoid redundancy with risk factors that were already considered´

We thank the reviewer for the additional insights as the authors strongly agree with this vision. For a matter of transparency of the search strategy described in  Figure 1, the authors believe that it would be more accurate to comment and cite only those studies that are emerging from the declared search criteria.

  1. By pointing out the current research gaps and biases, the review underscores the need for improved clinical decision-making tools. Better molecular markers and risk classifications will enable more personalized and effective treatment strategies for endometrial cancer patients. The review's reproduction of analysis for the confirmation cohort adds robustness to its findings. Reproducing and validating previous research is a critical step in confirming the reliability of biomolecular classifications and their clinical relevance. Highlighting the substantial concern of bias in data interpretation due to study limitations, the review calls for a more critical evaluation of existing research. This scrutiny is necessary to advance the field and ensure that clinical practices are based on sound evidence. The discussion on the evolution of biomolecular classification in endometrial cancer is insightful. Understanding how classifications have developed over time helps contextualize current challenges and directs efforts toward more effective future classifications.

We believe the reviewer is right therefore the abstract was rewritten in a more incisive way, while conclusions now have the comments that we previously included in the discussion section.

Round 2

Reviewer 1 Report

Comments and Suggestions for Authors

nonÄ™

Reviewer 4 Report

Comments and Suggestions for Authors

The manuscript has been revised according to the reviewer's comments. Please re-check the reference 9 and 21, since the journal citation (volume number) is similar. 

Reviewer 5 Report

Comments and Suggestions for Authors

accept